# *Paulownia* Organs as Interesting New Sources of Bioactive Compounds

**DOI:** 10.3390/ijms24021676

**Published:** 2023-01-14

**Authors:** Natalia Sławińska, Joanna Zając, Beata Olas

**Affiliations:** Department of General Biochemistry, Faculty of Biology and Environmental Protection, University of Lodz, Pomorska 141/3, 90-236 Lodz, Poland

**Keywords:** biological activity, phenolic compound, phytocompounds, oxidative stress, *Paulownia*

## Abstract

*Paulownia* spp. is a genus of trees in the Paulowniaceae family. It is native to southeastern Asia (especially China), where it has been cultivated for decorative, cultural, and medicinal purposes for over 2000 years. Depending on taxonomic classification, there are 6 to 17 species of *Paulownia*; *P. tomentosa*, *P. elongata*, *P. fortunei*, and *P. catalpifolia* are considered the most popular. Nowadays, Paulownia trees are planted in Asia, Europe, North America, and Australia for commercial, medical, and decorative purposes. Lately, growing interest in Paulownia has led to the development of various hybrids, the best-known being Clone in vitro 112, Shan Tong, Sundsu 11, and Cotevisa 2. Paulownia Clone in vitro 112 is an artificially created hybrid of two species of *Paulownia*: *P. elongata* and *P. fortunei*. The present review of selected papers from electronic databases including PubMed, ScienceDirect, and SCOPUS before 15 November 2022 describes the phytochemical characteristics, biological properties, and economic significance of various organs from different *Paulownia* species and hybrids, including *P. tomentosa*, *P. elongata*, *P*. *fortunei*, and Paulownia Clone in vitro 112. Many compounds from Paulownia demonstrate various biological activities and are promising candidates for natural preparations; for example, the leaves of Clone in vitro 112 have anti-radical and anticoagulant potential. However, further in vivo studies are needed to clarify the exact mechanism of action of the active substances and their long-term effects.

## 1. Introduction

*Paulownia* is a genus of trees in the Paulowniaceae family [1]. It is native to southeastern Asia (especially China), where it has been cultivated for decorative, cultural, and medicinal purposes for over 2000 years [1,2]. It is also known as the princess tree, royal tree, Kiri tree, empress tree, and phoenix tree [3,4], while its Chinese name is 泡桐 (pāo tóng) [5]. The genus *Paulownia* is believed to comprise 6 to 17 species depending on taxonomic classification [6]. Of these, *P. tomentosa*, *P. elongata*, *P. fortunei,* and *P. catalpifolia* are considered the most popular [4].

Nowadays, Paulownia trees are planted in Asia, Europe, North America, and Australia for commercial, medical, and decorative purposes [7]. Due to their fast growth rate and adaptability, they are considered invasive species in some countries. However, most of the risks could be attenuated by planting hybrids that produce infertile seeds (e.g., Clone in vitro 112) [1,6]. Paulownia can adapt to varied environmental conditions, and it has a fast growth rate and exceptional regenerative abilities; a cut tree trunk can regrow up to 2–4 m in one year [7]. In fact, it is one of the fastest-growing trees in the world, being able to produce several times more biomass in one year than some of the slower-growing species. These properties have led to an increased interest in establishing Paulownia plantations for the purpose of biomass production [1]. The use of Paulownia as a bioenergy crop, i.e., for the production of biofuel and CO_2_ sequestration is also being considered [8]. In addition, its ability to withstand high concentrations of heavy metals (e.g., Mn, Pb, or Zn) can be utilized in the rejuvenation of contaminated soil and reforestation [4,9]. Paulownia wood is also used in construction and to make paper pulp, furniture, and musical instruments [1,9,10], while the flower nectar serves as a source of high-quality honey [2,8,11]. The leaves have high protein content (approximately 20%) and can be used to make high quality, cheap, animal feed [11,12].

Trees from the genus *Paulownia* contain many phytochemicals with potential benefits for human health. C-geranylated flavonoids abundant in *P. tomentosa* are particularly interesting, as they can be found in a relatively small number of plant families. Examples include mimulone and diplacone [13,14]. Apart from C-geranylated flavonoids, Paulownia trees contain phenolic acids, phenolic glycosides, lignans, quinones, terpenoids, phytosterols, and glycerides [14,15,16,17,18]. They exhibit a wide range of biological activities, including antioxidant, anti-inflammatory, antibacterial, antiviral, neuroprotective, antiproliferative, anti-cancer, cytotoxic, and anti-hyperlipidemic activities (Figure 1) [14,19,20,21,22]. In China, Paulownia has been used as a traditional herbal medicine in the treatment of inflammatory bronchitis, tonsillitis, gonorrhea, traumatic bleeding, asthma, and hypertension [23]. Lately, increasing interest in Paulownia has led to the development of a range of hybrids, the best-known being Clone in vitro 112, Shan Tong, Sundsu 11, and Cotevisa 2 [1].

Paulownia Clone in vitro 112, also known as *Oxytree* for its large leaves and ability to absorb large amounts of CO_2_, is an artificially created hybrid of two species of *Paulownia*: *Paulownia elongata* and *Paulownia fortunei* [24]. The first plantations were established in 2014 [7]. The plant is believed to effectively improve air quality. It has a faster growth rate than other species of *Paulownia* and can tolerate a wide range of temperatures, which makes it a convenient and profitable biomass and bioenergy crop [25]. Since it has infertile seeds, planting Clone in vitro 112 does not carry any risks of it spreading uncontrollably and becoming invasive [6]. 

The present review describes the phytochemical characteristics, biological properties, and economic value of various Paulownia organs, including Paulownia Clone in vitro 112. This review is based on studies identified in electronic databases, including PubMed, ScienceDirect, and SCOPUS. The last search was run on 15 November 2022

## 2. Taxonomy of *Paulownia*

Although *Paulownia* currently belongs to the Paulowniaceae family, it was previously classified as a member of Scrophulariaceae [16]. The total number of *Paulownia* species is not universally agreed upon; depending on taxonomical classification, this number can range from 6 to 17 [6]. Li et al. [26] define eight species: *P. tomentosa*, *P. coreana*, *P. kawakamii*, *P. fortunei*, *P. elongata*, *P. catalpifolia*, *P. australis*, and *P. fargessi*. The Chinese Flora Editorial Committee does not accept *P. coreana* but lists two additional variations of *P. tomentosa*: *P. tomentosa* var. *tomentosa* and *P. tomentosa* var. *tsinlingensis* [5]. In addition, other authors have recognized *P. albipholea*, *P. taiwaniana*, and *P. glabrata* [8,27,28].

## 3. Botanical Characterization of *Paulownia*

Usually, a mature Paulownia grows to a height of 20–30 m; the tallest registered specimen was 50 m [3]. The trunk is typically around one meter thick but can reach two meters in suitable environmental conditions [6]. Root systems of Paulownia trees are well-developed and can grow up to a depth of 8 m. The upper section of the roots is densely packed, branched, and dichotomous. The bark is brown or black in color. Young specimens develop lenticels that later expand to form vertical cracks as the tree grows [1,3]. The leaves of a mature Paulownia are umbrella-shaped, 10–12 cm wide, and 15–30 cm long, with smooth, wavy edges. Younger trees have much larger leaves that can reach a width of 80 cm [29]. Its flowers bloom in May and June and have five petals that are white to light purple in color. The fruits are approximately 4 cm long and 2.5 cm wide. They ripen in autumn, each releasing up to 2000 winged seeds [3,30]. The surface of the leaves, fruits, and flowers is covered with glandular trichomes that secrete various substances [8,31,32].

Paulownia trees use C4 photosynthesis to fix carbon. Such plants are better adapted to a warm climate. In the right conditions, C4 plants can absorb approximately 10 times more CO_2_ than those that use C3 photosynthesis. C3 plants are less efficient at fixing carbon due to the poor specificity of the Rubisco enzyme, which binds oxygen and carbon dioxide. These inadequacies become more apparent at high temperatures and lower CO_2_ concentrations [7,29]. In contrast, C4 plants employ a mechanism that results in higher concentrations of CO_2_ at the catalytic sites of Rubisco, allowing them to function particularly well in environments with low atmospheric CO_2_ levels [33]. As a result, a single Paulownia tree can absorb 22 kg CO_2_ and release 6 kg O_2_ a year [3].

The optimal temperature for growth of Paulownia is 24 °C to 29 °C; however, it can survive in temperatures ranging from −18 °C to 35 °C. Despite this, it must be protected from sunburn and frostbite in locations that experience large seasonal differences in temperature. The optimum altitude for growth is 700–800 m above sea level, and it is most commonly found at latitudes between 40° N and 40° S [27]. It prefers wind-sheltered positions with high sunlight exposure [27,29]. To achieve optimal growth, mean annual rainfall should be between 800 and 2600 milliliters [34]. Paulownia can grow on peat and sandy soil with an optimal pH ranging from 5 to 8. Despite not thriving on clay or rocky soil, it has good adaptability—it can grow on salty (under 1%) or nutrient-poor soil, thanks to its ability to selectively absorb calcium and magnesium ions [27,34].

## 4. Commercial Uses and Economic Value of Paulownia

Fast growth rate and good adaptability are the reasons for increased interest in Paulownia as a biomass source. For example, it is a good candidate for short-rotation forestry (SRF). SRF is a type of tree cultivation where crops reach their optimal size and are ready to be harvested in 8 to 20 years [35]. Unlike Paulownia, most of the tree species commonly used in SRF plantations (e.g., poplar, willow, black locust, or alder) employ the less efficient C3 photosynthesis [36]. The rotation cycle of SPF is usually three to six years; trees that are planted more densely have shorter rotation times. It is often possible to plant up to 10,000 units per 1 ha [37]. In the past, the biomass collected from SRF plantations was used primarily to produce cellulose pulp; however, nowadays it is mainly utilized as a source of thermal or electrical energy, i.e., as a bioenergy crop [38]. In this aspect, biomass acquired from SRF could help replace fossil fuels as a source of energy and reduce the emissions of greenhouse gasses [39]. However, this form of tree cultivation has its disadvantages. SRF crops release a large quantity of volatile organic compounds (VOC) that contribute to tropospheric ozone production [37]. Tropospheric ozone is toxic—it can increase the production of reactive oxygen species in cells and impair CO_2_ absorption in plants [40,41].

Paulownia leaves have high nutritional value and are a good source of bioactive substances, which makes them a valuable animal feed component (Table 1) [10,42,43]. They are rich in minerals, proteins, nitrogen, and crude fiber. For example, *P. tomentosa* leaves have higher levels of manganese, zinc, tyrosine, and methionine than lucerne [42,44]. The addition of Paulownia leaves to rabbit feed (up to 15%) increased the blood concentration of high-density lipoprotein (HDL) and decreased the amount of low-density lipoprotein (LDL). Furthermore, it reduced the number of pathogenic bacteria in the caecum. However, at high concentrations, the rabbits demonstrated slower growth [44].

The leaves are also a good fertilizer and can enrich the soil with valuable organic matter and microorganisms. Fallen leaves support the growth of bacteria, which stimulate the production of various phytohormones, enzymes, biosurfactants, and precursors of secondary metabolites, which enhance plant growth and improve immunity to pathogens. These bacteria contribute to the circulation of minerals in the soil, bind atmospheric nitrogen, and decompose organic matter; however, they can also promote the occurrence of various diseases [7,45].

## 5. The Active Substances of *P. tomentosa*—The Most Extensively Studied Species of *Paulownia*

The most extensively studied species of *Paulownia* is *Paulownia tomentosa*. Studies have confirmed its anti-inflammatory, antioxidant, antibacterial, antiviral, and neuroprotective properties [46,47]. In traditional Chinese medicine, extracts from the wood, fruit, and bark were used to treat bronchitis, asthma, and bacterial infections [30]. *P. tomentosa* was shown to contain flavonoids, lignans, phenolic glycosides, quinones, terpenoids, phytosterols, and glycerides, as well as phenolic acids (p-hydroxybenzoic, gallic, vanillic, cinnamic, caffeic, and p-coumaric acids) [14,15,16,17,18] [Table 2].

Lately, interest in Paulownia has increased following the discovery of multiple geranylated flavonoids in *P. tomentosa*, many of which have never been isolated from any other plant. C-geranylated flavonoids, a group of flavonoid derivatives, are relatively rare and occur only in a small number of plant families [5,14]. They consist of a flavonoid skeleton and a monoterpenoid side chain [97]. The geranyl part is synthesized by the mevalonate pathway, while the flavonoid part is the product of the shikimic acid pathway. The two components are linked together by prenyltransferases [5]. C-geranylated flavonoids exhibit a wide range of activities, including antioxidant, anti-inflammatory, antibacterial, antiviral, antiparasitic, and cytotoxic activities [5,22,97,98]. Recent studies have shown that they could have the potential to be developed into anti-inflammatory drugs [17]. 

Among nine geranylated flavonoids isolated from the fruits of *P. tomentosa*, diplacone and 3′-O-methyl-5′-hydroxydiplacone demonstrated the most robust antioxidant activity [57]. In addition, mimulone and diplacone had anti-inflammatory effects (they reduced COX-2 activity), and diplacone downregulated the expression of tumor necrosis factor alpha (TNF-α) and monocyte chemoattractant protein 1 (MCP-1) [5]. A stem bark extract of *P. tomentosa* also had anti-inflammatory and antioxidant effects. It inhibited the influx of neutrophiles and macrophages, reduced interleukin-6 (IL-6) and TNF-α production, and decreased serum nitric oxide concentrations in a murine acute lung injury model. In addition, it inhibited nuclear factor-kappa B (NF-κB) activity and promoted superoxide dismutase 3 (SOD3) activation [91]. 

The C-genarylated flavanones from *P. tomentosa* fruits have demonstrated anti-inflammatory activity as well. Several compounds (diplacone, tomentodiplacone, tomentodiplacone N, mimulone H, and 3′,4′-O-dimethyl-5′-hydroxydiplacone) inhibited the expression of TNF-α by preventing IκB degradation. IκB degradation allows for transcription factor NF-κB to be translocated to the nucleus, where it activates the transcription of TNF-α [60]. Tomentodiplacone O inhibited the activity of cyclooxygenase COX-1, demonstrating higher selectivity for COX-1 than COX-2 compared to ibuprofen [17]. 

Several C-geranylated flavonoids isolated from *P. tomentosa* fruit have demonstrated antiproliferative and cytotoxic effects against the THP-1 cell line. Diplacone demonstrated the strongest activity in both regards, while 3′-O-methyl-5′-hydroxydiplacone exhibited a relatively strong antiproliferative, but weaker cytotoxic effect [14]. These results are in line with those of Zima et al., where diplacone and 3′-O-methyl-5′-hydroxydiplacone were found to be the most active C-geranylated flavonoids isolated from the fruits of *P. tomentosa* [57]. In addition, another geranylated flavonoid (CJK-7) upregulated autophagy and induced caspase-dependent cell death in the HCT-116 human colon carcinoma cell line [99].

A fruit extract inhibited the activity of protein tyrosine phosphatase 1B (PTP1B) and α-glucosidase, which are important targets in the treatment of obesity and diabetes. Geranylated flavonoids isolated from the extract also showed potent inhibitory activity. The most effective compound turned out to be mimulone [63].

Compounds isolated from *P. tomentosa* fruits demonstrated antiviral activity. They inhibited papain-like protease (PLpro) of Severe Acute Respiratory Syndrome Corona Virus 2 (SARS-CoV); the greatest activity was demonstrated by compounds that contained an unusual 3,4-dihydro-2H-pyran motif [100]. Antiviral activity was also observed in quinones isolated from the stem bark, more precisely, methyl-5-hydroxy-dinaphthol [1,2-2′,3′]furan-7,12-dione-6-carboxylate, which significantly reduced the cytopathic effects of type 1 and 3 polioviruses [89]. Moreover, *P. tomentosa* flower extract countered enterovirus (EV-71) infection, with the key substance being apigenin [48]. C-geranylated flavanones isolated from the fruits showed antibacterial activity against *Staphylococcus aureus* and several of its methicillin-resistant strains. Mimulone and 3′-*O*-methyldiplacol had the strongest effects, with MIC values ranging from 2 to 4 μg/mL [13].

Kim et al. reported that methanol extract from *P. tomentosa* flowers had neuroprotective properties. The extract reduced glutamate-induced toxicity in primary cultured rat cortical cells in a dose-dependent manner. Protection from glutamate-induced damage plays a crucial role in preventing neurodegenerative diseases. Glutamate is an endogenous amino acid that acts as an excitatory neurotransmitter. Although it plays a significant role in the nervous system by facilitating neuroplasticity, neuronal survival, and learning processes, it can also promote the development of neurodegenerative diseases such as Alzheimer’s disease, Parkinson’s disease, or epilepsy. Among five flavanones isolated from the extract (5,4′-dihydroxy-7,3′-dimenthoxy-flavanone, 5-hydroxy-7,3′,4′-trimenthoxyflavanone, diplacone, mimulone, and isoatriplicolide tiglate), isoatriplicolide tiglate had the most potent neuroprotective ability. Its incubation with rat cortical cells at concentrations of 1 µM and 10 µM improved cell viability to 43% and 78%, respectively [19]. 

In addition, geranylated flavonoids isolated from *P. tomentosa* fruits (6-geranyl-3,3′,5,5′,7-pentahydroxy-4′-methoxyflavane, diplacone, and 6-geranyl-3′,5,5′,7-tetrahydroxy-4′-methoxyflavanone) were able to mitigate the symptoms of Alzheimer’s disease by inhibiting the activity of acetylcholinesterase (AChE) and butyrylcholinesterase (BChE), resulting in increased concentrations of acetylcholine and butyrylcholine in the synapses. These are required for correct brain function, and it is thought that dysregulation of ACheE and BChE may lead to the progression of Alzheimer’s disease [101].

A furanquinone (methyl 5-hydroxy-dinaphtho [1,2-2′3′]furan-7,12-dione-6-carboxylate) isolated from the stem of *P. tomentosa* had an inhibitory effect on cathepsin K [90]. Cathepsin K is a protease expressed mainly in bone marrow, although small amounts can be also found in other tissues. It is involved in the process of bone matrix degradation by osteoclasts. Cathepsin K inhibitors are currently under evaluation as potential drugs for osteoporosis treatment [102].

## 6. *P. fortunei* and *P. elongate*—Predecessors of Paulownia Clone In Vitro 112

*P. fortunei* is also known as the Chinese parasol tree [97]. Its flowers are edible and can be used to make a dish called Zheng Cai. Extracts from different organs of the tree have also been used to treat bacterial infections such as dysentery, tonsillitis, and bronchitis, as well as enteritis and hypertension [20]. *P. elongata* demonstrates an above-average growth rate. In its second year, it can reach a height of 4 m and have a diameter of 5–6 cm [103].

*P. fortunei* flower extract is rich in flavonoids (including C-geranylated flavonoids), e.g., apigenin, luteolin, quercetin, kaempferol, β-sitosterol, mimulone, diplacone, hesperetin, thunberginol A, daucosterol, and their derivatives [104]. Its leaves contain mimulone, apigenin, luteolin, anserinoside, ursolic acid, maslinic acid, daucosterol, and beta-sitosterol [15,105,106]. Other compounds isolated from *P. fortunei* include phenylpropanoid glycosides, phenolic acids, triterpenes, and lignans [106,107,108] (Table 3).

Four C-geranyl flavonoids isolated from the flower extract have shown potent anti-inflammatory activity. They protected cardiomyocytes from lipopolysaccharide (LPS)-induced inflammation and decreased serum levels of IL-6 and TNF-α [22].

*P. fortunei* flower polysaccharide (PFFPS) is a water-soluble compound composed of 10 monosaccharides, mostly galactose (28.61%), rhamnose (18.09%), glucose (15.21%), and arabinose (15.91%). PFFPS was shown to improve cellular and humoral immunity. Chickens injected with PFFPS demonstrated increased leucocyte counts and higher IL-2 and IFN-γ concentrations [97].

Liu et al. report that *P. fortunei* flower extract decreased total cholesterol concentration in plasma, prevented hepatic lipid accumulation, and facilitated weight loss in mice fed with high-fat diets. HDL levels were increased, while plasma insulin and glucose concentrations were reduced. These effects can be attributed to the upregulation of 5′AMP-activated protein kinase (AMPK) pathway and the activation of insulin receptor substrates (IRS1). AMPK plays an important role in regulating lipid metabolism, while IRS1 activates the insulin signaling cascade. The phosphorylation levels of both AMPK and IRS1 were significantly increased in mice supplemented with the extract [20].

Extracts from fresh and fermented leaves of *P. fortunei* also had antibacterial properties. They inhibited the growth of bacteria (*Salmonella enterica*, *Streptococcus pyogenes*, *Staphylococcus aureus*, *Pseudomonas aeruginosa*, *Paenibacillus alvei*) and fungi (*Candida albicans*), although this inhibitory effect was more pronounced against Gram-negative bacteria [11].

Sheep fed with *P. elongata* leaves had lower leukocyte and erythrocyte counts and demonstrated lower plasma glucose concentrations [109].

Extracts from fresh and dry leaves of *P. fortunei* and *P. elongata* also appear to have antioxidant activity, as indicated by studies based on the TREAC assay (TROLOX Equivalent Antioxidant Capacity), in which the reactivity of an antioxidant is compared to the activity of TROLOX—a water-soluble vitamin E analog. The results are expressed as percentage inhibition of the ABTS^•+^ radical cation in comparison to TROLOX. Extracts from *P. fortunei* leaves showed a mean inhibition of 61.03% (fresh leaves) and 95.09% (dry leaves), while for *P. elongata*, these values were 50.21% and 60.88%, respectively [28,110,111]. The total flavonoid contents of the fresh leaf extracts were 157.53 µg/mL for *P. fortunei* and 102.58 µg/mL for *P. elongata* [28].

## 7. Paulownia Clone In Vitro 112—Characterization

Paulownia Clone in vitro 112 (also known as Oxytree and Biotree) is a hybrid of *P. elongata* and *P. fortunei.* It can withstand a wide range of temperatures (−25 °C to +45 °C), which allows it to be cultivated in many parts of the globe [7]. It is one of the fastest-growing deciduous trees in the world—it can reach up to 16 m in height and 35 cm in trunk diameter in only six years. After cutting, the trunk can regrow four to five times. Thanks these regenerative abilities, the wood can be harvested more than once. In addition, its root system can reach a depth of 9 m [25]. The tree can also rejuvenate contaminated soil and improve groundwater retention [112]. It absorbs large amounts of CO_2_, i.e., up to 111 tons/ha/year; in comparison, oak can assimilate only 9.1 tons/ha/year [25]. Another advantage of Oxytree is that it only produces infertile seeds, thus reducing the risk of it becoming an invasive species [6].

As wood harvested from Paulownia Clone in vitro 112 is lightweight and durable [6], it is sometimes referred to as ‘aluminum wood’; 1 m^3^ weighs approximately 310 kg and is rated as class I on the Janka scale, indicating a very soft wood [36]. The wood of Oxytree is lighter than the wood of other species of deciduous trees. Stochmal et al. [25] estimated that it is approximately 50% lighter, while Bikfalvi [112] reported that it is lighter by 30%. Moreover, Biotree wood is a good thermal insulator, has fine texture, and is resistant to deformation [112].

## 8. Chemical Content of Paulownia Clone In Vitro 112 and Its Biological Activity

Currently, there are few publications describing the chemical constituents of Paulownia Clone in vitro 112. As the chemical content varies between different species and cultivars, more research is needed to ascertain the content of secondary metabolites and their properties [4,113].

Adach et al. showed that the majority of active constituents of Oxytree leaves were phenolic compounds, the most predominant being verbascoside and its derivatives (methoxyverbascoside, hydroxyverbascoside). Other phenolics included apigenin-HexA-HexA, luteolin-HexA-Hex, and caffeic acid-Hex-dHexA. The total phenolic content was 205.5 mg·g^−1^ ± 6.41 [24,114]. Dżugan et al. reported that the total phenolic content of leaf tissue is 248.51 mg GAE/g (mg of gallic acid equivalents per gram of dry mass), and total flavonoid content is 147.71 mg QE/g (mg of quercetin equivalent per gram of dry mass) [4].

In addition, the leaf extract contained the iridoids catalpol and aucubin or 7-hydroxytomentoside, with the total iridoid content of 15.16 mg·g^−1^ ± 0.274. Out of these, aucubin and its isomer 7-hydroxytomentoside were present in the highest concentrations. The extract also contained a small number of triterpenoids, including C_30_H_48_O_6_-Hex, C_30_H_48_O_5_, maslinic acid, and C_30_H_48_O_3_. Total triterpenoid content was 3.65 mg·g^−1^ ± 0.278 [66]. The substances presented in Table 4 were isolated by Adach et al. in 2020 and 2021.

Verbascoside is a phenylethanoid glycoside [66] produced via the shikimic acid pathway [115]. It has shown antioxidant, anti-inflammatory, antibacterial, neuroprotective, antitumoral, photoprotective, gastroprotective, wound-healing, and anti-osteoporotic properties [66,115] and could inhibit the activity of cytochrome P (CYP) enzymes. Verbascoside has been isolated from several plants used in traditional Chinese medicine, including *Verbenaceae* and *Plantago* [66].

Aucubin has anti-inflammatory, antioxidant, hepatoprotective, pancreas-protective, hypolipidemic, and antibacterial activities [74,116]. It reduced lipid peroxidation and increased the activity of antioxidant enzymes (i.e., catalase, glutathione peroxidase, and superoxide dismutase) in the liver of rats with streptomycin-induced diabetes. It could lower the blood glucose concentration as well [116]. Moreover, aucubin prevented steroid-induced bone loss by upregulating AMPK-dependent autophagy, which averted osteoblast apoptosis [117].

Catalpol can penetrate the blood brain barrier and has anti-inflammatory, antioxidant, anti-apoptosis, antitumor, and neuroprotective properties [75,76]. It is used to treat age-related macular degeneration, which is a disease manifested by visual distortions, dark spots, and impaired central vision [118].

Maslinic acid is a pentacyclic triterpene that has gathered attention due to its numerous beneficial properties and pharmacological safety. It was reported to have antioxidant, anticancer, anti-inflammatory, antidiabetogenic, and antiviral abilities [86] (Table 4).

**Table 4 ijms-24-01676-t004:** Substances isolated from Paulownia Clone in vitro 112, their biological activity, and their presence in other species of *Paulownia*.

Paulownia Clone In Vitro 112			Presence in Other *Paulownia* Species
Substance Type	Substance	Biological Activity	
Phenolic compounds	Verbascoside	Antioxidant, anti-inflammatory, neuroprotective, antiproliferative, muscle atrophy relieving, wound healing [66]	*P. tomentosa*: stem bark, fruits *P. tomentosa var. tomentosa*: bark *P. coreana*: bark, leaves [8,19,56,91,119]
	Apigenin	Antioxidant, anti-inflammatory, antiviral, apoptosis-inducing, anti-depression, antidiabetic, anti-amyloidogenic [48,49]	*P. tomentosa:* bark, flowers *P. fortunei*: flowers, leaves *P. coreana:* bark [23,46,105,107,120]
	Luteolin	Anti-inflammatory, neuroprotective, anti-cancer [61,62]	*P. tomentosa*: bark, flowers, fruits, leaves *P. fortunei*: flowers, leaves *P. coreana:* bark [23,59,105,107,120,121]
	Caffeic acid	Antioxidant, anti-inflammatory, antimicrobial inhibiting melanin production, cytostatic, vasorelaxant, anti-angiogenic, anti-atherosclerotic [122,123]	*P. tomentosa:* flowers, leaves *P. coreana:* bark, fruits [120,121,124]
Iridoids	Aucubin	Antioxidant, anti-inflammatory, hepatoprotective, pancreas-protective, hypolipidemic, antibacterial, bone loss prevention [74]	*P. tomentosa*: leaves [16,114]
	7-Hydroxytomentoside		*P. tomentosa:* bark, leaves, roots *P. coreana* [16,125]
	Catalpol	Antioxidant, anti-inflammatory, neuroprotective, anti-apoptosis, antitumor [75,76]	*P. tomentosa* *P. coreana* [125]
Triterpenoids	Maslinic acid	Antioxidant, anti-inflammatory, antidiabetogenic, anticancer, antiviral [86]	*P. tomentosa*: leaves [15]

Other substances isolated from the leaves and twigs of Paulownia Clone in vitro 112 include small amounts of dicaffeoylacteoside, 1-O-caffeoyl-6-O-alpha-rhamnopyranosyl-beta-glycopyranoside, 3-(4-Hydroxyphenyl)-1,2-propanediol4′-O-glucoside, campneoside I, acetyl acteoside (tubuloside B), epimeredinoside A, and didehydroxyacteoside [43]. Dicaffeoylacteoside showed good radical-scavenging activity against 1,1-diphenyl-2-picrylhydrazyl (DPPH) radical (SC_50_ = 19.6 ± 1.4 μM) and was cytotoxic toward SK-LU-1, MCF7, HepG2, and HeLa cancer cell lines [126,127]. Tubuloside B demonstrated neuroprotective properties. At the doses of 1, 10, and 100 mg/mL, it attenuated TNF-α-mediated apoptosis in SH-SY5Y neuronal cells [128]. A similar effect was observed in PC12 neuronal cells; 5–100 μg/mL of tubuloside B protected the cells from apoptosis induced by MPP^+^, an active metabolite of 1-methyl-4-phenyl-1,2,3,6-tetrahydropyridine (MPTP) [129].

Oxytree leaf extract had antioxidant activity. Adach et al. tested the effect of the extract and four fractions (A, B, C, and D) on human plasma treated with H_2_O_2_/Fe. Fractions A–C contained mostly verbascoside and its derivatives, as well as apigenin diglucuronide and luteolin diglucuronide. Fraction A and B contained iridoids. Fraction D contained phenolics, mainly acetylverbascoside and dimethylverbascoside, as well as apigenin and luteolin. Both the extract and all of the fractions significantly inhibited lipid peroxidation and oxidation of plasma protein thiol groups at the two highest concentrations (10 and 50 μg/mL). Moreover, fractions C and D were able to inhibit carbonylation of plasma proteins at all tested concentrations (1, 5, 10, and 50 μg/mL) [114].

Dżugan et al. have reported that the extracts from Oxytree leaves had lower antioxidant and antibacterial activity than extracts from other tested clones, with *P. tomentosa* x *P. fortunei* clones demonstrating the strongest effects. Additionally, the leaf blade extract showed four to nine times greater biological activity than the petiole extract. Higher activity correlated with a higher polyphenol concentration and a greater share of flavonoids in the polyphenol fraction (which ranged from 60 to 86% in the majority of cases). *P. elongata* x *P. fortunei* had the lowest antioxidant activity and polyphenol and flavonoid contents out of all tested clones [4].

The extract and four fractions (A–D) from the leaves of Clone in vitro 112 also showed anti-platelet activity, successfully inhibiting ADP-induced platelet aggregation at the highest concentration (50 μg/mL). They also lowered the adhesion of thrombin-activated platelets to fibrinogen and collagen. Lipid peroxidation was reduced in thrombin-activated platelets at all tested concentrations (1, 5, 10, and 50 μg/mL), although these results were not always statistically significant. The strongest effect was observed with fraction D; at the concentration of 50 μg/mL, the peroxidation was reduced by 60%. All the preparations increased the concentration of O_2_^−^. in resting and activated platelets. Overall, the extract had stronger antiplatelet activity than the fractions [24]. Moreover, fraction D showed the strongest anticoagulant activity in whole blood, which was determined with the Total Thrombus-Formation Analysis System (T-TAS) [113].

## 9. Conclusions

The reviewed literature demonstrated that the trees of *Paulownia* genus produce many promising chemical compounds (e.g., verbascoside, diplacone, mimulone, apigenin, catalpol, aucubin, and maslinic acid) with various biological activities (e.g., antioxidant, anti-inflammatory, antiproliferative, antibacterial, antiviral, neuroprotective, and hepatoprotective activities). Extracts and fractions from various Paulownia organs show beneficial properties as well. For example, the leaves of P. Clone in vitro 112 had anti-radical and anticoagulant effects, making them potential candidates for natural preparations (Figure 1). However, there is a need for more studies that would clarify the exact mechanisms of action and determine which active compounds are responsible for these effects. Moreover, animal studies and clinical trials should be performed to determine the in vivo efficiency of preparations and compounds from Paulownia and check their long-term effects and safety.

## Figures and Tables

**Figure 1 ijms-24-01676-f001:**
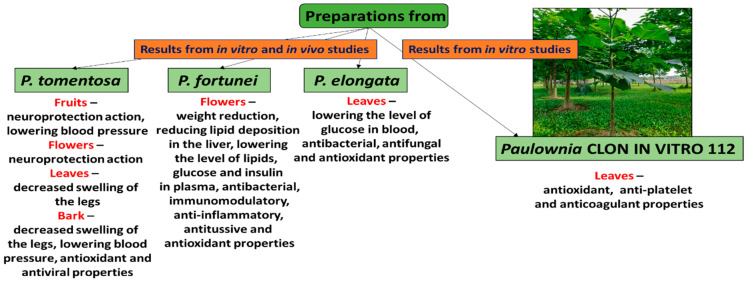
Biological activity of preparations from organs of various species of *Paulownia*.

**Table 1 ijms-24-01676-t001:** Components of Paulownia leaves ([3,27], modified).

Component	Quantity
Organic matter	91.4%
Protein	22.6%
Nitrogen	2.8–3.0%
Calcium	2.1%
Zinc	0.9%
Iron	0.6%
Phosphorus	0.6%
Potassium	0.4%

**Table 2 ijms-24-01676-t002:** Active substances isolated from different parts of *P. tomentosa*.

Location in Plant	Type of Chemical Compound	Active Substances	Biological Activity	Citations
Flowers	Flavonoids	Apigenin	Antioxidant, anti-inflammatory, antiviral, proapoptotic, anti-depression, antidiabetic, anti-amyloidogenic	[46,48,49]
		Apigenin-7-*O*-β-D-glucoside	Antioxidant, antidiabetic	[46,50,51]
		Quercetin	Antioxidant, anti-inflammatory, antimicrobial, anti-hypertensive, hypoglycemic, anti-hyperlipidemic, antitumor	[46,52,53]
		Quercetin-3-*O*-β-D-glucoside	Antioxidant, anti-inflammatory, anti-allergic, anti-glycation, antidiabetic, chemopreventive, cardioprotective	[46,54]
		3′-Methoxyluteolin-7-*O*-β-D-glucoside		[46]
		Tricin-7-*O*-β-D-glucopyranoside	Antihypoxic	[46,55]
Fruits	Polyphenols	Diplacone	Antioxidant, anti-inflammatory, antiproliferative, anti-cancer	[5,56,57,58,59,60]
		3′-*O*-Methyl-5′-hydroxydiplacone	Antioxidant, antiproliferative	[56,57,58,59,60]
		Luteolin	Anti-inflammatory, neuroprotective, anti-cancer	[56,57,58,59,60,61,62]
		Mimulone	Anti-inflammatory, antidiabetic, antibacterial, neuroprotective	[5,13,56,57,58,59,60,63]
		Schizolaenone C		[56,57,58,59,60]
		Sesamin	Antioxidant, anti-inflammatory, anti-asthmatic, anticancer, hepatoprotective, nephroprotective, hypotensive, anti-atherosclerotic, cardioprotective, anti-diabetic, anticancer	[56,57,58,59,60,64]
		Tomentodiplacone	Antibacterial	[56,57,58,59,60,65]
		Tomentomimulol		[56,57,58,59,60]
		Verbascoside	Antioxidant, anti-inflammatory, neuroprotective, antiproliferative, muscle atrophy relieving, wound healing	[56,57,58,59,60,66]
		Isoverbascoside	Antioxidant, anti-inflammatory, neuroprotective, nephroprotective, anti-glycation	[67,68,69,70,71]
Leaves	Iridoids	7-β-Hydroxyharpagide		[72,73]
		Aucubin	Antioxidant, anti-inflammatory, hepatoprotective, pancreas-protective, hypolipidemic, antibacterial, bone loss prevention	[72,73,74]
		Catalpol	Antioxidant, anti-inflammatory, neuroprotective, anti-apoptosis, antitumor	[72,73,75,76]
		Paulownioside		[72,73]
		Tomentoside		[72,73]
		7-Hydroxytomentoside		[72,73]
	Phytosterols	β-sitosterol	Antioxidant, anti-inflammatory, antibacterial, anti-amyloid β, anti-hyperlipidemic, immunomodulatory, modulation of gut microbiota, anticancer, antidiabetic, cardioprotective, hepatoprotective, neuroprotective	[15,77,78,79,80,81,82,83,84]
		Daucosterol	Antioxidant, anti-inflammatory, immunomodulatory, anticancer, neuroprotective, antidiabetic, hypolipidemic	[15,85]
	Triterpenoids	Maslinic acid	Antioxidant, anti-inflammatory, antidiabetic, anticancer, antiviral	[15,86]
		Pomolic acid	Anticancer, hypotensive, prevention of platelet aggregation, antiviral	[15,87]
		Ursolic acid	Anticancer, hepatoprotective, hypotensive, cardioprotective, neuroprotective, improvement of muscle mass, anti-osteoclastogenesis	[15,88]
		3-Epiursolic acid		[15]
Stem and bark	Quinones	Methyl 5-hydroxy-dinaphtho [1,2-2′3′]furan-7,12-dione-6-carboxylate	Cathepsin K inhibition	[89,90]
	Phenolic compounds	Apigenin	Antioxidant, anti-inflammatory, antiviral, apoptosis-inducing, anti-depression, antidiabetic, anti-amyloidogenic	[23,48,49,91]
		Campneoside II	Anti-complement	[23,91,92]
		Castanoside F		[23,91]
		Ellagic acid	Antioxidant, anti-inflammatory, antidiabetic, anti-hypercholesterolemic, cardioprotective, neuroprotective, hepatoprotective, anticancer, antibacterial, antiviral	[23,91,93]
		Isocampneoside II	Antioxidant, anti-complement, inhibition of aldose reductase	[23,91,92,94,95]
		Luteolin	Anti-inflammatory, neuroprotective, anti-cancer	[23,61,62,91]
		Verbascoside	Antioxidant, anti-inflammatory, neuroprotective, antiproliferative, muscle atrophy relieving, wound healing	[23,66,91]
		Isoverbascoside	Antioxidant, anti-inflammatory, neuroprotective, nephroprotective, anti-glycation	[23,67,68,69,70,71,91]
Wood	Lignans	Sesamin	Antioxidant, anti-inflammatory, anti-asthmatic, anticancer, hepatoprotective, nephroprotective, hypotensive, anti-atherosclerotic, cardioprotective, anti-diabetic, anticancer	[10,64,96]
		Paulownin		[10,96]

**Table 3 ijms-24-01676-t003:** Substances isolated from flowers and leaves of *P. fortunei*.

Location	Substance	Citation
Flowers	β-Sitosterol	[97,104,107,108]
	Abscisic acid	
	Apigenin	
	Apigenin-7-*O*-beta-D-glucoside	
	Arbutin	
	Daucosterol	
	Diplacone	
	3′-*O*-Methyldiplacol	
	3′-*O*-Methyldiplacone	
	Hesperetin	
	Homoeriodictyol	
	Kaempferol-3-*O*-beta-D-glucoside	
	Kaempferol-7-*O*-beta-D-glucoside	
	Luteolin	
	Luteolin-7-*O*-beta-D-glucoside	
	3′-Methoxyluteolin-7-*O*-beta-D-glucoside	
	Mimulone	
	Naringenin-7-*O*-beta-D-glucoside	
	*P. fortunei* flower polysaccharide (PFFPS)	
	Quercetin-3-*O*-beta-D-glucoside	
	Quercetin-7-*O*-beta-D-glucoside	
	Ursolic acid	
	1-Acetoxy-3-hydroxypropan-2-yl-3-hydroxypentanoate	
	4-Hydroxybenzyl-beta-D-glucoside	
	5,7,4′-Trihydroxy-3′-methoxy flavone	
	6-Geranyl-3,3′,5,7-tetrahydroxy-4′-methoxyflavanone	
Leaves	β-Sitosterol	[105,106]
	Anserinoside	
	Apigenin	
	Daucosterol	
	Luteolin	
	Maslinic acid	
	Mimulone	
	Pomolic acid	
	Ursolic acid	
	3α-Hydroxyl-ursolic acid	

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
