# Peer review of "Paulownia Organs as Interesting New Sources of Bioactive Compounds"

_ijms, 2023, doi:10.3390/ijms24021676_

Round 1

Reviewer 1 Report

Many articles on Paulownia have been published recently. In this way, I consider that the manuscript has great potential for publication.

I have some considerations to make in order to contribute to this review.

Abstract: presents a lot of information similar to the introduction, I suggest making it more succinct and contemplating a general approach to the manuscript.

Table 2 - I suggest inserting a column to cite the references.

Table 4 - I suggest quoting in the text or as a footnote that the substances presented in the table were isolated by Adach et al. 2022 and 2021, without having to cite this reference in each substance presented.

In the first paragraph below Table 4, the authors could mention whether or not there are studies on the bioactive properties of the compounds isolated by Puchalska et al. (2021).

I consider it important to quote the manuscript:

CHO, J. K. et al. Geranylated flavonoids displaying SARS-CoV papain-like protease inhibition from the fruits of Paulownia tomentosa. Bioorganic and Medicinal Chemistry, v. 21, n. 11, p. 3051–3057, 2013. Disponível em: <http://dx.doi.org/10.1016/j.bmc.2013.03.027>.

More emphasis may also be given to antimicrobial properties, which despite few studies yet, the following manuscript brings important information:

NAVRÁTILOVÁ, Alice; SCHNEIDEROVÁ, Kristýna; VESELÁ, Daniela; HANÁKOVÁ, Zuzana; FONTANA, Anna; DALL’ACQUA, Stefano; CVAÄŒKA, Josef; INNOCENTI, Gabbriella; NOVOTNÁ, Jana; URBANOVÁ, Marie; PELLETIER, Jerry; Č͎EK, Alois; ŽEMLIÄŒKOVÁ, Helena; ŠMEJKAL, Karel. Minor C-geranylated flavanones from Paulownia tomentosa fruits with MRSA antibacterial activity. Phytochemistry, vol. 89, p. 104–113, 2013. https://doi.org/10.1016/j.phytochem.2013.01.002.

Author Response

Reviewer 1

Many articles on Paulownia have been published recently. In this way, I consider that the manuscript has great potential for publication.

Thank you for reviewing the manuscript and providing such helpful comments. All of them have been taken into consideration when revising the manuscript.

I have some considerations to make in order to contribute to this review.

Abstract: presents a lot of information similar to the introduction, I suggest making it more succinct and contemplating a general approach to the manuscript.

Response: We have corrected the Abstract.

Table 2 - I suggest inserting a column to cite the references.

Response: We have inserted a column with citations

Table 4 - I suggest quoting in the text or as a footnote that the substances presented in the table were isolated by Adach et al. 2020 and 2021, without having to cite this reference in each substance presented.

Response: We have corrected this. We have added in the text: “The substances presented in the Table 4 were isolated by Adach et al. 2020 and 2021”. We have removed these references in Table 4.

In the first paragraph below Table 4, the authors could mention whether or not there are studies on the bioactive properties of the compounds isolated by Puchalska et al. (2021).

Response: We have added information about the activity of some of these compounds: “Dicaffeoylacteoside showed good radical-scavenging activity against 1,1-diphenyl-2-picrylhydrazyl (DPPH) radical (SC50 = 19.6 ± 1.4 μM) and was cytotoxic toward SK-LU-1, MCF7, HepG2, and HeLa cancer cell lines [97,98]. Tubuloside B demonstrated neuroprotective properties. At the doses of 1, 10, and 100 mg/mL, it attenuated TNF-α-mediated apoptosis in SH-SY5Y neuronal cells [99]. Similar effect was observed in PC12 neuronal cells. 5-100 μg/mL of tubuloside B protected the cells from apoptosis induced by MPP+, an active metabolite of 1-methyl-4-phenyl-1,2,3,6-tetrahydropyridine (MPTP) [100].”

I consider it important to quote the manuscript:

CHO, J. K. et al. Geranylated flavonoids displaying SARS-CoV papain-like protease inhibition from the fruits of Paulownia tomentosa. Bioorganic and Medicinal Chemistry, v. 21, n. 11, p. 3051–3057, 2013. Disponível em: <http://dx.doi.org/10.1016/j.bmc.2013.03.027>.

Response: This article was included in the review: “Compounds isolated from P. tomentosa fruits demonstrated antiviral activity. They inhibited papain-like protease (PLpro) of Severe Acute Respiratory Syndrome Corona Virus 2 (SARS-CoV); the greatest activity was demonstrated by compounds that contained an unusual 3,4-dihydro-2H-pyran motif [71].”

More emphasis may also be given to antimicrobial properties, which despite few studies yet, the following manuscript brings important information:

NAVRÁTILOVÁ, Alice; SCHNEIDEROVÁ, Kristýna; VESELÁ, Daniela; HANÁKOVÁ, Zuzana; FONTANA, Anna; DALL’ACQUA, Stefano; CVAÄŒKA, Josef; INNOCENTI, Gabbriella; NOVOTNÁ, Jana; URBANOVÁ, Marie; PELLETIER, Jerry; Č͎EK, Alois; ŽEMLIÄŒKOVÁ, Helena; ŠMEJKAL, Karel. Minor C-geranylated flavanones from Paulownia tomentosa fruits with MRSA antibacterial activity. Phytochemistry, vol. 89, p. 104–113, 2013. https://doi.org/10.1016/j.phytochem.2013.01.002.

Response: We have added information about antibacterial properties: “C-geranylated flavanones isolated from the fruits showed antibacterial activity against Staphylococcus aureus and several of its methicillin-resistant strains. Mimulone and 3’-O-methyldiplacol had the strongest effects, with MIC values ranging from 2 to 4 μg/mL [72].”

Reviewer 2 Report

The manuscript entitled " Paulownia organs as new interesting sources of bioactive compounds"" is a good review of the work on the plant. However, I have some major concerns about the writing part of the review article. The article appears to have been written so fast. There are many grammatical errors which should be avoided. The use of commas and semicolons may improve readability. Hence, after a major modification of the article it can be considered for publication.

1.      Thorough editing is required both in terms of language and format. Make sure that the font type and font size of the text are unique throughout the manuscript.

2.      The abstract section has shortcomings. The present abstract is not justifying the title of this review paper. There is no information on the bioactive compounds from Paulownia spp. I suggest to follow the pattern in an unstructured manner like "a brief background on the genus Paulownia, main objectives of the review, main findings of the review, conclusion and future perspectives".

3.      The authors have written the hybrid, Paulownia Clone in vitro 112 in different ways in the manuscript. Please follow a unique pattern.

4.      In the Introduction section it is stated that “However, due to their fast growth rate and adaptability, in some countries they are considered invasive species (Jakubowski et al. 2018)”. So being an invasive plant how can you justify the propagation of these plants? Being invasive, do you recommend these spp. considering their medicinal value alone?

5.      The authors have not mentioned any compounds from Paulownia spp in the Introduction part also. You have just explained some economic aspects and pharmacological value only. Authors must include the chemical components of different Paulownia spp in detail.

6.      In Table 2, include the nature of activity of each compound also.

7.      Italicize the scientific names of plants throughout the manuscript.

8.      The authors must elaborate the conclusion section giving more emphasis on the bioactive compounds from Paulownia spp and their therapeutic value. Authors suggested to perform more in vivo studies. Is it enough?  What about the molecular mechanism of action, clinical trials?  

9.      Check and format references according to the journal guidelines.

Author Response

Reviewer 2

The manuscript entitled " Paulownia organs as new interesting sources of bioactive compounds"" is a good review of the work on the plant. However, I have some major concerns about the writing part of the review article. The article appears to have been written so fast. There are many grammatical errors which should be avoided. The use of commas and semicolons may improve readability. Hence, after a major modification of the article it can be considered for publication.

  1. Thorough editing is required both in terms of language and format. Make sure that the font type and font size of the text are unique throughout the manuscript.

Response: We have revised the text in terms of language formatting.

  1. The abstract section has shortcomings. The present abstract is not justifying the title of this review paper. There is no information on the bioactive compounds from Paulownia spp. I suggest to follow the pattern in an unstructured manner like "a brief background on the genus Paulownia, main objectives of the review, main findings of the review, conclusion and future perspectives".

Response: We have corrected the Abstract.

  1. The authors have written the hybrid, Paulownia Clone in vitro 112 in different ways in the manuscript. Please follow a unique pattern.

Response: We have corrected this.

  1. In the Introduction section it is stated that “However, due to their fast growth rate and adaptability, in some countries they are considered invasive species (Jakubowski et al. 2018)”. So being an invasive plant how can you justify the propagation of these plants? Being invasive, do you recommend these spp. considering their medicinal value alone?

Response: We have added an explanation: “Due to their fast growth rate and adaptability, they are considered invasive species in some countries. However, most of the risks could be attenuated by planting hybrids that produce infertile seeds (e.g., Clone in vitro 112) [1,6].”

  1. The authors have not mentioned any compounds from Paulownia spp in the Introduction part also. You have just explained some economic aspects and pharmacological value only. Authors must include the chemical components of different Paulownia spp in detail.

Response: We have added the chemical components of Paulownia to the introduction “Trees from the genus Paulownia contain many phytochemicals with potential benefits for human health. C-geranylated flavonoids abundant in P. tomentosa are particularly interesting, as they can be found in relatively small number of plant families. Examples include mimulone and diplacone [13,14]. Apart from C-geranylated flavonoids, Paulownia trees contain phenolic acids, phenolic glycosides, lignans, quinones, terpenoids, phytosterols, and glycerides [14–18].”

  1. In Table 2, include the nature of activity of each compound also.

Response: We have added the activities of the compounds.

  1. Italicize the scientific names of plants throughout the manuscript.

Response: We have corrected plant names.

  1. The authors must elaborate the conclusion section giving more emphasis on the bioactive compounds from Paulownia spp and their therapeutic value. Authors suggested to perform more in vivo Is it enough? What about the molecular mechanism of action, clinical trials? 

Response: The reviewed literature demonstrated that the trees of Paulownia genus produce many promising chemical compounds (e.g., verbascoside, diplacone, mimulone, apigenin, catalpol, aucubin, or maslinic acid) with various biological activities (e.g., antioxidative, anti-inflammatory, antiproliferative, antibacterial, antiviral, neuroprotective, or hepatoprotective). Extracts and fractions from various Paulownia organs show beneficial properties as well. For example, the leaves of P. Clone in vitro 112 had anti-radical and anticoagulant effects, making them potential candidates for natural preparations [Figure 1]. However, there is a need for more studies that would clarify the exact mechanisms of action and determine which active compounds are responsible for these effects. Moreover, animal studies and clinical trials should be performed to determine the in vivo efficiency of preparations and compounds from Paulownia and check their long-term effects and safety.”

  1. Check and format references according to the journal guidelines.

Response: We have checked and formatted the references.

Round 2

Reviewer 2 Report

The authors have revised the manuscript considerably well